# Reproductive Resumption in Winter and Spring Related to *MTNR1A* Gene Polymorphisms in Sarda Sheep

**DOI:** 10.3390/ani12212947

**Published:** 2022-10-26

**Authors:** Maria Consuelo Mura, Giovanni Cosso, Luisa Pulinas, Vincenzo Carcangiu, Sebastiano Luridiana

**Affiliations:** Department of Veterinary Medicine of Sassari, University of Sassari, Via Vienna 2, 07100 Sassari, Italy

**Keywords:** melatonin, reproductive recovery, male effect, fertility rate, Sarda breed sheep

## Abstract

**Simple Summary:**

Small ruminants at the Mediterranean latitudes show a reproductive seasonality determining the seasonal availability of their production, resulting in problems for the marketing of these products. In order to better manage this crucial aspect, it is essential to study the genes that can influence seasonality. In different sheep breeds, polymorphisms at the melatonin receptor 1A (*MTNR1A*) gene are associated with an early reproductive recovery in spring, which is advantageous to shorten the anestrus period. Nevertheless, there are still shortcomings in the relationship between these polymorphisms and the reproductive activity, and if these mutations could influence the effectiveness of the ram introduction in the flock, based on different time periods, succeeding the seasonal anestrous. The present research evaluated the impact of two single nucleotide polymorphisms (SNP) rs430181568 and rs407388227 at the *MTNR1A* gene exon 2 on the resumption of reproductive activity in 240 ewes carrying different genotypes at these loci and submitted to different periods of ram introduction (February, March, April and May) in order to provide new information to stakeholders. Animals were divided into four groups (A–D) based on genotype and month of ram introduction. Fertility rate, distance from ram introduction to lambing and litter size were recorded in order to measure potential differences in crucial reproductive parameters based on different genotypes. The results confirmed the association between SNPs and reproductive recovery, with beneficial effects even in February and March, which are immediately subsequent to the photo-refractoriness period.

**Abstract:**

The aim of the present research was to evaluate the association between the SNPs rs430181568 and rs407388227 located in the *MTNR1A* gene with the reproductive recovery of Sarda sheep in different months of ram introduction in the flock (February, March, April and May). In order to address this research gap, we selected two farms, each of which consisted of approximately 1000 animals; a total of 800 ewes (400 for each farm) were genotyped for the two single nucleotide polymorphisms rs430181568 and rs407388227 located in the exon 2 of the *MTNR1A*. These SNPs are completely linked; thus, each genotype of rs430181568 corresponded to the same genotype for rs407388227. Among the genotyped animals, 240 individuals were selected and divided into four homogeneous groups (A, B, C and D) of 60 subjects, each group based on age (range 3–6 years old), body condition score (BCS) (range 2.0–4.0) and genotype (20 ewes carrying CC/CC, 20 CT/CT and 20 TT/TT genotype). The dates of the ram introduction in each group were 15 February, 15 March, 15 April and 15 May, respectively. In all groups, the lambing date and the number of lambs born from 150 to 220 days after the ram introduction were recorded. In all the groups, the genotypes CC/CC and CT/CT of the polymorphism (rs430181568 and rs407388227) showed the greatest fertility (the ratio between the number of lambed ewes and the ewes exposed to the rams) (*p* < 0.01) and the shortest distance between ram introduction to lambing (*p* < 0.01), compared with the TT/TT genotype. In conclusion, we determined that the polymorphisms rs430181568 and rs407388227 were associated with reproductive recovery, after ram introduction, even in February and March, months subsequent to the photorefractoriness period.

## 1. Introduction

Reproduction in sheep is largely regulated by photoperiodic changes through seasonal variations in the daily duration of night and day, which results in an annual alternation of anestrous and reproductive seasons [1,2]. In ewes, at temperate latitudes, ovulatory activity is suppressed for several months in spring and summer during the anestrous period [3]. The changes in photoperiod affect circulating melatonin levels, causing higher melatonin concentrations during short-day photoperiods. This increase in melatonin secretion coincides with the seasonal reproductive resumption in Mediterranean sheep [4].

However, if, on the one hand, the reproductive seasonality allows the lambing during the best season for the growth of newborns, on the other, it results in a seasonal availability of the productions with considerable problems for their marketing.

Furthermore, the study of genes that can influence seasonality in order to manage the genetic potential of animals better is of considerable importance. The melatonin (MT) receptors play a role in the process played by this hormone in controlling reproductive seasonality [5,6,7]. Among all, the receptor type MT1 showed a direct association with the regulation of reproductive activity in sheep [8]. The MT1 receptor is encoded by the *MTNR1A* gene, which is currently recognized as a candidate gene related to the seasonal reproductive activity in sheep [9,10,11]. Some of the polymorphic sites found within the *MTNR1A* gene have been shown to improve reproductive performance in general and to affect seasonal reproductive resumption in spring in different sheep breeds [6,12,13].

The reproductive resumption in spring usually occurs after the sensitization period of about 35 days and long days, also known as the photorefractory period, after the winter solstice, and this reproductive resumption is a gradual event [14,15]. This fact suggests that sheep pass from a state of deep to shallow anestrus, which allows the animal a gradual transition toward reproductive recovery [16]. In previous research, we hypothesized that the SNPs rs430181568 and rs407388227 located in the exon 2 of the *MTNR1A* gene are associated with the effect of the photoperiod and, therefore, showed an earlier spring reproductive recovery [13,17]. In our previous studies on the Sarda breed sheep, different SNPs were found in exon 1 and 2, but only those mentioned above in exon 2 showed an association with reproductive activity [18]. Therefore, the aim of the present study was to verify if the animals carrying the different genotypes of these two polymorphisms show a different reproductive recovery after the photorefractoriness period (February and May). Then, the resumption of reproductive activity was evaluated after the introduction of the rams in the flock in February, March, April and May of the sheep carrying the different genotypes of the polymorphisms at position rs430181568 and rs407388227 of the *MTNR1A* gene.

## 2. Materials and Methods

The animal care conditions and management practices agreed with the procedures stated by the organization in charge of Animal Welfare and Experimentation (OPBSA) of the University of Sassari. All the animals were under the control of the National Health Veterinary Service of Italy, following the guidelines of the Animal Welfare Act and the codes of recommendations for the welfare of livestock of the Italian Ministry of Health.

### 2.1. Animals and Experimental Design

In the present study, two sheep farms located in North Sardinia (on the 40° N) were selected. The farms were under the same pedoclimatic condition and had the same feeding and sanitary management, as they were served by the same veterinarian and nutritionist. During the day, the ewes grazed on green pasture (leguminous and gramineous grasses) and also received a concentrate supplement (300 g per head/daily of commercial food—crude protein 20.4% and 12.5 MJ ME/kg DM) when they returned from pasture at sundown, during the afternoon milking. During the night, they were shepherded into a pen, where they had free access to hay (crude protein 11.1% and 7.2 MJ ME/kg DM) and water. The total number of animals raised in each farm was approximately 1000 Sarda sheep, but a total of 800 sheep (400 per farm) were selected for genotyping analysis of the SNPs (single nucleotide polymorphism) rs430181568 and rs407388227. Of the 800 animals, 240 were selected on the basis of their alleles/genotypes so that they could be divided into groups of approximately the same age and BCS. The SNPs rs430181568 and rs407388227 are completely linked (D’ = 1 and r^2^ = 1); thus, the frequency of genotypes and alleles are the same. Furthermore, the frequency by farm were: first farm, CC/CC (0.63), CT/CT (0.19) and TT/TT (0.17), whilst second farm CC/CC (0.60), CT/CT (0.21) and TT/TT (0.19).

Among these animals, on the 15th of January, we chose 120 animals from each farm, and afterward, they were kept on one farm. On the same date, we divided them into 4 homogeneous groups (A, B, C and D). Each group was based on the date of the previous lambing (from 15 October to 15 November), the farm of origin (30 animals for each group), age (range 3–6 years old), BCS (body condition score, range 2.0–4.0) and genotype (20 ewes carrying CC/CC, 20 CT/CT and 20 TT/TT genotype). The groups are organized as described in the following Table 1:

All the ewes were identified both by ear tag number and rumen boluses code in order to avoid recognition errors. Individual BCS was recorded according to the methods of Russel [19], with scores ranging from 1 (very poor condition) to 5 (very good condition) in half-unit increments. The same person assigned the scores by touching the amounts of muscling and fat deposition in the loin region.

### 2.2. Blood Sampling and Genotyping

At the beginning of the study, an individual blood sample was drawn from the jugular vein to extract DNA. Blood was collected using sterile vacuum tubes (BD Vacutainer System, Belliver Industrial Estate, Plymouth, UK) with ethylenediamine tetraacetic acid (EDTA) as an anticoagulant. The DNA was extracted from 200 µL of whole blood using a genomic DNA extraction kit (NucleoSpin^®^ Blood, Macherey-Nagel, Dueren, Germany). The polymerase chain reaction (PCR) method was carried out to amplify the whole exon 2 of the *MTNR1A* gene. An amount of 150 ng of the obtained genomic DNA was subjected to PCR, using primers Fw: 5′–GGC CCT AAC CCA TGT TTT CT–3′ and Rv: 5′–CTC CCA CTC TGT TCC CTG AA–3′, delimiting an 1157 base pairs (bp) fragment, corresponding to the entire exon 2 and partial 3′UTR, within the latest ovine genome version (ARS-UI_Ramb_v2.0—GenBank Assembly Accession Number: GCF_016772045.1). PCR reaction was carried out in 50 µL final volume, containing 1X PCR Buffer (minus MgCl_2_) (20 mM Tris-HCl (pH 8.0), 40 mM NaCl, 2 mM Sodium Phosphate, 0.1 mM EDTA, 1 mM DTT, stabilizers, 50% (*v*/*v*) glycerol), 1.2 mM of MgCl_2_, 200 μM of each dNTP, 0.4 μM of each primer, 1.25 units (U) of Taq DNA polymerase (HOT FIREPol^®^ Polymerase, Solis BioDyne, Tartu, Estonia) and ultrapure water DNase/RNase free (Water PCR grade, Solis BioDyne, Tartu, Estonia) up to 50 μL. PCR conditions included an initial activation of the Hot Start DNA Polymerase by a 15 min incubation step at 95 °C followed by an initial denaturation at 94 °C for 3 min and by subsequent 35 cycles shared in denaturation at 94 °C for 1 min, annealing at 54 °C for 1 min, extension at 72 °C for 1 min and, finally, a final extension at 72 ◦C for 5 min, on MAXYGENE II Thermal Cycler (Axygen^®^ Tewksbury, MA, USA). Ten microliters of PCR products were electrophoresed at 80 V for 30 min in a 1.5% ultrapure agarose gel (*w*/*v*) (iNtrRon Biotechnology, Sangdaewon-Dong, South Korea) added with 7 μL of RedSafe stain (iNtrRon Biotechnology, Sangdaewon-Dong, South Korea) in 1X TBE electrophoresis buffer, and visualized by ultraviolet transillumination (UVItec, Cambridge, UK), together with a 100-bp Ladder (GeneRuler, Thermo Scientific™, Waltham, MA, USA). 

The PCR products were purified using the kit Multiscreen^®^ filter plates (Millipore, Merk Life Science S.r.l., Milan, Italy) and then sequenced in forward and reversed direction by a commercial service. Before the Sanger sequencing (ABI PRISM 3730 DNA Analyzer, Applied Biosystems), the samples were prepared using the Big Dye Terminator sequencing kit v3.1 (Applied Biosystems). The alignment of the resulting sequences and the comparison with the latest version of the sheep genome—ARS-UI_Ramb_v2.0—GenBank Assembly Accession Number: GCF_016772045.1–was performed using the BLAST program (www.ncbi.nlm.nih.gov/blast/ (accessed on 30 July 2022). The BioEdit Sequence Alignment Editor software, selecting the IUB weight matrix (for DNA) scoring matrix, was used for the sequence alignments [20]. The Chi-square test was used to evaluate the deviation from Hardy–Weinberg equilibrium and Maf (minor allele frequency) was calculated as the ratio between the less frequent allele and the total number of alleles screened.

### 2.3. Reproductive Data Collection

A veterinarian evaluated the clinical health and welfare of the animals included in the study. In each group, fertile rams were introduced in different periods. The rams (Sarda breed) were four years old, and they had already produced offspring in the previous mating seasons. Furthermore, a veterinarian performed a genital examination and did not find any abnormalities in the reproductive system. The rams of the two farms were kept in a group to facilitate management. Before introducing rams in each group, in order to obtain the optimal efficacy from the male effect, rams were previously isolated from ewes for 90 days, during which a distance more than 1500 m between the two sexes was maintained to avoid that sound, sight or smell could invalidate reproductive results. In each group, there were three rams, so the ratio between males and females was 1:20. The chosen rams carried CC/CC genotype in order to not affect the reproductive results. Rams were introduced on February 15 in group A, on March 15 in group B, on April 15 in group C and on May 15 in group D. Furthermore; each group was housed in paddocks distant approximately 4 km from each other in order to avoid visual, olfactory and sound signals. Rams and ewes remained together for 70 days in each group. In order to obtain more detailed information on the reproductive activity, the rams were provided with marker harnesses that were replaced every 10 days; thus, mating was tracked every day. Gestation was diagnosed in all ewes every week from 45 days after the rams’ introduction to 45 days after the rams’ removal. Individual reproductive performance of the ewes in each group was evaluated by recording the date of lambing and the number of newborn lambs in order to assess the fertility rate (the ratio between the number of lambed ewes and the ewes exposed to the rams) and the litter size (the number of lambs born in a litter). Finally, the mean distance in days from ram introduction to lambing (DRIL) was calculated for each ewe in each group.

### 2.4. Statistical Analysis

All the statistical analyses were conducted using R statistical software (Version 4.1.2 R Core Team 2021 R: A language and environment for statistical computing. R Foundation for Statistical Computing, Vienna, Austria; https://www.R-project.org/ (accessed on 10 August 2022). The additional packages were ‘lme4′ [21], ‘emmeans’ [22] and ‘agricolae’ [23].

For the analysis of potential influences on fertility (lambing/not lambing), a Chi-square test was used. The differences in litter size and DRIL among periods of ram introduction and genotypes were analyzed using the following model:Y_ijk_ = μ + T_i_ + B_j_ + (T_i_B_j_) + + BCS_k_ + e_ijk_,(1)
where Y_ijk_ is the trait measured for each animal (DRIL or litter size), T_i_ is the fixed effect of the period of ram introduction (4 levels, i = February, March, April, May), B_j_ is the fixed effect of the genotype (3 levels, j = CC, CT, TT), BCS_k_ is the effect of the BCS and e_ijk_ is the random residual effect of each observation. DRIL and litter size was expressed as least square means ± SEM. Multiple comparisons of the least square means were performed using Tukey’s method. The Chi-square test was used to evaluate the deviation from Hardy–Weinberg equilibrium, Maf (minor allele frequency) was calculated as the ratio between the less frequent allele and the total number of alleles screened. All the results were considered statistically significant when the *p*-value was below 0.05.

## 3. Results

The electrophoresis migration pattern consisted of a genomic fragment of 1157 bp that covers the exon II of the *MTNR1A* gene. The sequencing showed nine polymorphisms highlighted in following Table 2:

As found in our other previous studies, the polymorphisms rs430181568 and rs407388227 were always linked [24]. The statistical analysis showed an association between the genotypes rs430181568 and rs407388227 and the reproductive performances in Sarda adult ewes. In fact, the ewes carrying CC/CC or CT/CT genotype showed the greatest fertility in all the periods of ram introduction (*p* < 0.05) (Table 3). There were no abortions detected amongst any of the ewes. In groups A and B, no return to the estrus cycle was recorded, while in groups C and D, two and three ewes, respectively. In contrast, the different periods of ram introduction did not show statistical differences.

Regarding the litter size, no statistical differences were found considering the variable’s genotype and period of ram introduction (mean 1.15 ± 0.03). However, the genotypes showed an association with the DRIL: the genotypes CC/CC and CT/CT displayed the shortest distance in days from ram introduction to lambing (*p* < 0.01), compared to the T/T ewes (Table 4).

## 4. Discussion

This study analyzed the exon 2 of the *MTNR1A* gene in Sarda ewes, confirming, by sequencing, the nine mutations we formerly found in other research [18]. Previously, we also analyzed exon 1, but no polymorphism was found there. Furthermore, we studied the relationship between the found polymorphisms in exon 2 with reproductive activity in Sarda and in other sheep breeds, and only rs430181568 and rs407388227 showed a statistically significant association [7,13,18,24,25]. Thus, as these topics were extensively discussed, in this study, we focused only on the rs430181568 and rs407388227 SNPs. The allelic and genotype frequency of the found polymorphisms were similar to those that were published in previous research in the Sarda sheep breed [24]. The two mutations in position rs430181568 and rs407388227 were completely linked, and these data agree with what was found in previous research on different breeds of sheep [6,13,26]. Ewes with the CC/CC and CT/CT genotypes of these two polymorphisms showed better reproductive performance in all four observed periods than those with the TT/TT genotype. Indeed, sheep with the genotypes CC/CC and CT/CT show a greater number of lambing than the TT/TT in all four periods observed. The association between these polymorphisms of the *MTNR1A* gene with reproductive activity agrees with what was found in previous research on different breeds of sheep [6,9]. This association could be due to the action that melatonin performs at the ovarian level. In fact, in the ovary, the receptors for melatonin were identified in the granulosa cells and in the corpus luteum, and the effects in improving the growth of the follicle and the functioning of the corpus luteum were highlighted [27,28]. At the ovarian level, receptors are also involved in influencing ovarian activity, and MT1 seems to be the one that most influences this activity [29]. Indeed, the silencing of this receptor has led to a greater expression of apoptotic genes and a decrease in anti-apoptosis genes in granulosa cells [29]. However, the silencing of the MT1 receptor in granulosa cells does not totally block the effect of melatonin in regulating apoptosis, suggesting another pathway of control presumably mediated through the MT2 receptor [30]. Therefore, these two receptors can collaborate with each other to ensure that the ovarian follicle can develop adequately through the maintenance of granulosa cells [29]. Thus, it can be hypothesized that in our study, these effects mentioned above on the ovarian follicle allowed for greater fertility in sheep with the CC/CC and CT/CT genotypes compared to those with TT/TT. In sheep with the TT/TT genotype, the ovarian follicles that formed, in large part, could have undergone apoptosis and for which ovulation did not occur, and therefore neither did pregnancy.

Melatonin is considered to be the scavenger of free radicals and a stimulator for the production of antioxidant enzymes [31,32,33]. These effects are evident in the granulosa cells, where melatonin stimulates the expression of some antioxidant genes [34]. Moreover, in this case, the silencing of MT1 reduces the expression of antioxidant genes, and the action of melatonin is also lost, thus indicating that MT1 regulates the antioxidant effect of melatonin at the level of the ovaric follicle, safeguarding its development [29,30].

Furthermore, by binding to its receptors, melatonin influences the development of the corpus luteum and the secretion of progesterone by granulosa cells [29,35]. This stimulatory effect on progesterone secretion was demonstrated on corpus luteum cells of pregnant sows and is presumably due to the increase in two mediators (cholesterol side-chain cleavage enzyme) (P450scc) and steroidogenic acute regulatory (StAR)) of the biosynthesis of the progesterone [36]. Furthermore, it was shown that this effect is mainly mediated by the MT1 receptor, even if a certain effect is also carried out by the MT2 one.

The two SNPs, rs430181568 and rs407388227, as reported in several studies on both the Sarda and other breeds [13,25], are associated with the regulation of reproductive seasonality. The SNP rs407388227 causes an amino acid change (Val > Ile) at position 220 of the protein chain (GenBank access number AAB17721.1). This aa change is located in the fifth transmembrane domain (TM5), which is crucial for the functionality of the MT1 receptor [37]. Missense mutations within this domain lead to important changes in signal transmission [38,39]. Furthermore, significant differences in cAMP inhibition were observed between sheep with Val220 and Ile220 [40], thus suggesting a possible change in melatonin signal transmission in sheep carrying different alleles of the SNP rs407388227. These variations in the perception of the melatonin signal could have influenced all the aforementioned actions at the ovarian level, thus determining the best reproductive performance recorded in this study in sheep with the CC/CC and CT/CT genotypes compared to sheep carrying the TT/TT genotype. In fact, with greater protection of the follicle, better functioning of the corpus luteum generated by a better transmission of the melatonin signal in animals carrying the CC/CC and CT/CT genotypes should be the basis for obtaining a better reproduction of these sheep.

However, in February and March, the fertility recorded in the CC/CC and CT/CT genotypes was higher than that recorded in the months of April and May, but the differences were not statistically evident. On the other hand, the fertility rate in sheep carrying the TT/TT genotype did not show variations in relation to the month of observation. The development of the ovarian follicle is a complex process involving endocrine, paracrine and autocrine mechanisms. The primordial follicles must grow and pass from the primary, pre-antral and antral stages before reaching the preovulatory stage and being able to release an oocyte that can undergo fertilization. Interestingly, despite there being a large number of follicles developed during each reproductive cycle, just a few of these will mature and be able to ovulate. Consequently, all the so-called follicles selected for growth are doomed to death due to atresia. These selected follicles are the largest and have follicular fluid in their follicular antrum, which contains water, electrolytes, serum proteins and high concentrations of steroid hormones secreted by granulosa cells [41]. Furthermore, large follicles have a higher melatonin concentration in the follicular fluid than small follicles [42]. The presence of melatonin and its precursors and of the two enzymes synthesizing melatonin, NAT and HIOMT, has been documented in the ovary [43]. Therefore, the ovary is able to produce melatonin, but most of that present in the follicular fluid is thought to be derived from the bloodstream [44]. Thus, it is logical to think that when melatonin circulation is high, that present in the ovarian follicle is also high and carries out its activities through its receptors. Therefore, for all the aforementioned actions, it is presumed that the best reproductive performances obtained in the months of February and March in animals with the genotype CC/CC and CT/CT in our study are linked to the higher levels of melatonin found in those months compared to the months of April and May. Furthermore, it can be hypothesized that animals carrying these two genotypes are less affected by the photoperiod and, therefore, photo refractivity has a less strong effect than animals carrying the TT/TT genotype. Furthermore, animals with the CC/CC and CT/CT genotypes show a much shorter distance in days between the introduction of the males and the birth than the carriers of the TT/TT genotype. Therefore, it can be assumed that animals with the CC/CC and TT/TT genotypes are in a very superficial anestrus stage, and only the male effect is able to re-trigger reproductive activity compared to animals with the TT/TT genotype. Furthermore, at the latitude of Sardinia, the lambing in spring allow a good production of milk just for a few months because the hot summer climate considerably restrains its production. Therefore, to overcome this problem and also to exploit the growth cycle of natural green grass, various reproduction management techniques are implemented so as to allow the mating of sheep in spring with offspring in autumn [45].

Another finding, which was concurrent with our previous studies, was that the litter sizes in Sarda breed sheep were not significantly affected by genotype [18]. Considering that Sarda is a dairy sheep, it could be assumed that litter size, not being a key trait for farmers, constantly remain near the typical mean values of this breed, regardless of genotype.

In summary, the two analyzed genotypes are able to improve the reproductive recovery between the end of winter and the beginning of spring, guaranteeing that the dairy sheep, such as the Sardinian one, have early deliveries in autumn and consequently a longer lactation. Furthermore, with the targeted use of these animals carrying different genotypes, it could be thought of as a de-seasonalizing reproduction, and therefore have milk production throughout the year.

## 5. Conclusions

We concluded that the polymorphisms rs430181568 and rs407388227 of the *MTNR1A* gene are associated with variance in fertility rate and in DRIL in four months observed. Furthermore, sheep with the CC/CC and CT/CT genotypes show earlier reproductive recovery compared to animals with the TT/TT genotypes. Therefore, animals with genotypes CC/CC and CT/CT appear to be less affected by photoperiod changes and therefore have a shorter photo-refractoriness period than TT/TT animals. These findings highlight the significant association of these polymorphisms with the reproductive recovery in Sarda breed sheep and give a useful tool for improving genetic selection. Therefore, this fact could be exploited through reproduction plans aimed at obtaining reproduction all year round and consequently also a production of milk throughout the year. Therefore, further studies must be carried out to clarify the actual role played by the polymorphisms of the *MTNR1A* gene in the transmission of the melatonin signal at the follicular level.

## Figures and Tables

**Table 1 animals-12-02947-t001:** Groups (month of ram introduction), number of animals per group, genotypes per group, number of animals per genotype, average of age and BCS for each genotype inside of the groups.

Group n. 60	Farm	Genotype	n.	Average Age	Average BCS
A	1	CC/CC	10	4.8 ± 0.6	2.9 ± 0.8
		CT/CT	10	4.7 ± 0.6	3.0 ± 0.6
		TT/TT	10	4.8 ± 0.7	2.8 ± 0.6
	2	CC/CC	10	5.0 ± 0.7	2.9 ± 0.6
		CT/CT	10	4.9 ± 0.7	3.0 ± 0.5
		TT/TT	10	4.9 ± 0.5	2.8 ± 0.6
B	1	CC/CC	10	4.9 ± 0.7	3.1 ± 0.7
		CT/CT	10	4.9 ± 0.5	2.8 ± 0.7
		TT/TT	10	4.8 ± 0.5	3.0 ± 0.8
	2	CC/CC	10	4.9 ± 0.6	3.0 ± 0.6
		CT/CT	10	5.0 ± 0.8	2.9 ± 0.5
		TT/TT	10	4.9 ± 0.4	2.9 ± 0.7
C	1	CC/CC	10	4.8 ± 0.6	2.9 ± 0.
		CT/CT	10	4.9 ± 0.4	2.8 ± 0.7
		TT/TT	10	4.9 ± 0.6	3.0 ± 0.5
	2	CC/CC	10	5.0 ± 0.6	2.9 ± 0.6
		CT/CT	10	4.9 ± 0.5	2.9 ± 0.5
		TT/TT	10	4.9 ± 0.7	2.9 ± 0.7
D	1	CC/CC	10	4.7 ± 0.5	2.9 ± 0.8
		CT/CT	10	5.0 ± 0.7	2.9 ± 0.5
		TT/TT	10	4.9 ± 0.4	2.8 ± 0.7
	2	CC/CC	10	4.9 ± 0.6	3.0 ± 0.5
		CT/CT	10	4.9 ± 0.6	3.0 ± 0.6
		TT/TT	10	4.8 ± 0.8	2.9 ± 0.5

**Table 2 animals-12-02947-t002:** Identification code of the variant, position in the current assembly, genotypes, genotype frequencies in all the 800 animals, minor allelic frequency (MAF), Hardy–Weinberg equilibrium, amino acid changes and SIFT (Sorting Intolerant From Tolerant) score (prediction of the effect of an amino acid change on protein function).

Id Variant	Position in Oar_Rambouillet_v1.0	Position in ARS-UI_Ramb_v2.0	GENOTYPE	Genotype Frequency	Allele	MAF	HW Equilibrium	AA Change	SIFT
rs419680097	17355611	15571482	CC	0.28	A	0.47	0.971	-	
			CA	0.50					
			AA	0.22					
rs406779174	17355458	15571329	GG	0.51	A	0.29	0.952	-	
			GA	0.41					
			AA	0.08					
rs430181568	17355452	15571323	CC	0.62	T	0.28	0.000	-	
			CT	0.20					
			TT	0.18					
rs407388227	17355358	15571229	CC	0.62	T	0.28	0.000	Ile/Val	0.26
			CT	0.20					
			TT	0.18					
rs404378206	17355190	15571061	CC	0.53	T	0.27	0.883	Ile/Val	0.05
			CT	0.40					
			TT	0.07					
rs429718221	17355173	15571044	GG	0.12	G	0.31	0.203	-	
			GA	0.37					
			AA	0.51					
rs403212791	17354971	15570842	GG	0.85	A	0.07	0.417	Cys/Arg	0.07
			GA	0.15					
			AA	0.00					
rs426523476	17354963	15570834	GG	0.11	G	0.30	0.341	-	
			GA	0.38					
			AA	0.51					
rs413084140	17354943	15570814	CC	0.52	T	0.30	0.153	His/Arg	0.13
			CT	0.36					
			TT	0.12					
rs403826495	17354935	15570806	CC	0.52	T	0.29	0.440	Ile/Val	1
			CT	0.38			0.971		
			TT	0.10					

**Table 3 animals-12-02947-t003:** Fertility rate in each level of the two variables of the study (period of ram introduction and genotype rs430181568 and rs407388227).

		Genotypes		
Period	CC/CC	CT/CT	TT/TT	*p* Value
February	0.84	0.85	0.61	0.000
March	0.86	0.85	0.59	0.000
April	0.83	0.84	0.59	0.000
May	0.82	0.83	0.58	0.000
*p* value	0.186	0.094	0.978	

The statistical difference among genotypes is shown in each row, while the statistical difference among different periods of ram introduction is shown in each row.

**Table 4 animals-12-02947-t004:** Least square means ± SEM of DRIL and *p*-value for distance in days from ram introduction to lambing (DRIL) of each level of the fixed effects (genotype, period of ram introduction and their interaction).

Factor	Level	DRIL	*p* Value
Genotype	CC/CC	173.08 ± 7.78	0.0001
	CT/CT	173.60 ± 8.02	
	TT/TT	187.15 ± 8.06	
Period	February	177.38 ± 9.79	0.986
	March	177.10 ± 10.57	
	April	177.71 ± 9.59	
	May	177.60 ± 10.84	
Genotype by Period	CC/CC–February	172.00 ± 8.08	0.973
	CC/CC–March	172.88 ± 6.88	
	CC/CC–April	173.53 ± 6.18	
	CC/CC–May	174.07 ± 10.31	
	CT/CT–February	173.75 ± 9.20	
	CT/CT–March	174.46 ± 6.36	
	CT/CT–April	172.60 ± 7.87	
	CT/CT–May	173.60 ± 8.96	
	TT/TT–February	186.78 ± 8.19	
	TT/TT–March	187.76 ± 8.43	
	TT/TT–April	188.16 ± 7.18	
	TT/TT–May	186.00 ± 9.03	

## Data Availability

The data presented in this study are available on request from the corresponding author. The data are not publicly available to preserve the privacy of the data.

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
