# Peer review of "Reproductive Resumption in Winter and Spring Related to MTNR1A Gene Polymorphisms in Sarda Sheep"

_animals, 2022, doi:10.3390/ani12212947_

Round 1
Reviewer 1 Report
Review: Reproductive resumption in winter and spring related to 2 MTNR1A gene polymorphisms in Sarda sheep. Animals 2022
Results of this study show that polymorphism at the 2nd exon of the MTNR1A gene is associated with out-of-season fertility of Sarda sheep, following exposed to ram effect. This is a well design experiment with interesting results. However, more clarification should be given before the manuscript can be recommended for publication.
Comments:
Line 25: Each of the farms had about 1000 ewes or together they had 1000 ewes?
Line 36: The polymorphism was associated with… not influencing. There is no evidence for direct or indirect influence.
Line 99: the frequency of the CC, CT and TT animals among the 800 tested ewes, by farm, should be reported.
Line 99: Genotyping was carried out for 800 or 1600 ewes?
Line 101: it is not clear whether the experiment was carried out in the two farms (120 ewes in each farm) or all the 240 ewes where grouped in one farm.
Line 105: it is not clear how the different genotypes in each group (A, B, C, D) where distributed between the two farms. Please clarify.
Line 127: write µl.
Line 164: How rams` fertility was evaluated?
Line 168: Saying that there were 12 rams (line 190), did the rams where switched between farms?
Line 168: What was the genotype of the rams? (it may affect the results).
Line 168: I may understand that rams of group A, after serving for 70 days where moved to Group C in each farm. Is that correct? Did the rams were moved back for 20 days before introducing to group C?
Line 168: How far from each other were the paddocked of groups A, B. C, D? One may assume that if they were close enough, the ram effect in group A has some influence of groups B,C,D. This should be clarified.
Line 172: Does birthweight of lambs was recorded? If so, birthweight of lambs, according to litter size, should be reported. This is in light of the suggestion that melatonin seasonal fluctuation may affect lambs’ birth weight (Gootwine and Rosov, 2006: Seasonal effects on birth weight of lambs born to prolific ewes maintained under intensive management. Livestock Science 105: 277–283).
Line 190: statistical model: If three rams were introduced to a group, and there was no individual hand mating, but group mating, how a specific ram can be included as an effect in the model?
Line 200: please give in Table 2 the frequency of the different mutation in the 800 samples.
Line 213: Please provide Means and SD for prolificacy of Sarda sheep in this study.
Tale 4: Please use capital letters for all months and factors
Line 224: Rephrase.
Line 264: There is no evidence that the two mutations are involved. They are associated….
Line 275: The effect on fertility was not significant (Table 3).
Line 312: the conclusion is that the polymorphism is associated with variance in fertility. No evidence is for “direct improvement”.
Line 323: Supporting information is not available on the net.
Author Response
Results of this study show that polymorphism at the 2nd exon of the MTNR1A gene is associated with out-of-season fertility of Sarda sheep, following exposed to ram effect. This is a well design experiment with interesting results. However, more clarification should be given before the manuscript can be recommended for publication.
Dear reviewer,
Thank you for the revision, your suggestions greatly improved the understanding of our manuscript.
Listed below there are our responses.
Comments:
Rev1#point 1: Line 25: Each of the farms had about 1000 ewes or together they had 1000 ewes?
Author’s response#point1: thank you for giving us the opportunity to clarify the number of the research: each farmed consisted of 1000 ewes approximately, and we genotyped a total of 800 individuals (approximately 400 per farm). We specified this in the text
Rev1#point2: Line 36: The polymorphism was associated with… not influencing. There is no evidence for direct or indirect influence.
Author’s response#point2: we changed “influence” with “were associated”
Rev1#point3: Line 99: the frequency of the CC, CT and TT animals among the 800 tested ewes, by farm, should be reported.
Author’s response#point3: We added the frequency of each genotype by farm
Rev1#point:4 Line 99: Genotyping was carried out for 800 or 1600 ewes?
Author’s response#point4: we genotyped 800 ewes (approximately 400 per farm) because we wanted to constitute homogeneous groups considering the genotype, the age of the animals, the BCS and the previous lambing. In the text we specified we genotyped 800 ewes
Rev1#point5: Line 101: it is not clear whether the experiment was carried out in the two farms (120 ewes in each farm) or all the 240 ewes where grouped in one farm.
Author’s response#point5: the experiment was carried out in all the 240 ewes grouped in one farm (120 ewes per farm), as suggested, we specified this in the text
Rev1#point6: Line 105: it is not clear how the different genotypes in each group (A, B, C, D) where distributed between the two farms. Please clarify.
Author’s response#point6. We chose 120 animals per farm in order to obtain groups with balanced qualitative variables (both genotype and farm). Following your suggestion, we added the number of animals of each farm per group and we updated Table 1 accordingly.
Rev1#point7: Line 127: write µl.
Author’s response#point7: we added “µl”
Rev1#point8: Line 164: How rams` fertility was evaluated?
Author’s response#point8: Thank you for pointing out this lack: the rams used in the present research were adult (4 years) and their fertility has been proven by the registration of their offspring in the past seasons. Furthermore, a veterinarian performed a physical examination of genitals in order to assess their health conditions.
Rev1#point9: Line 168: Saying that there were 12 rams (line 190), did the rams where switched between farms?
Author’s response#point9: Thank you for giving the opportunity to clarify: in each group (n.60) there were 3 rams, for a total of 12 animals, 6 rams from each farm.
Rev1#point10: Line 168: What was the genotype of the rams? (it may affect the results).
Author’s response#point10: The rams chosen were genotyped and they were all CC/CC. We added this information in the text.
Rev1#point11: Line 168: I may understand that rams of group A, after serving for 70 days where moved to Group C in each farm. Is that correct? Did the rams were moved back for 20 days before introducing to group C?
Author’s response#poin ramst11: In our experimental design we had 12 rams divided in 4 group (3 rams per group). All the rams remained into the assigned group for 70 days, without switching among groups.
Rev1#point12: Line 168: How far from each other were the paddocked of groups A, B. C, D? One may assume that if they were close enough, the ram effect in group A has some influence of groups B,C,D. This should be clarified.
Author’s response#point12: Each group was housed in separate paddocks distant approximately 4Km from each other, in order to avoid visual, olfactory, and sound signals. We added this additional information in the text.
Rev1#point13: Line 172: Does birthweight of lambs was recorded? If so, birthweight of lambs, according to litter size, should be reported. This is in light of the suggestion that melatonin seasonal fluctuation may affect lambs’ birth weight (Gootwine and Rosov, 2006: Seasonal effects on birth weight of lambs born to prolific ewes maintained under intensive management. Livestock Science 105: 277–283).
Author’s response#point13: in the Sardinian sheep farming system, milk production is the main purpose; thus, the lambs’ birthweight is not useful for farmers. In fact, lambs remain with their mother for 30 day, and they are suckled ad libitum. After this time period lambs are removed and mothers started their real production period, being milked daily. For these reasons the lambs birthweight was not recorded.
Rev1#point14: Line 190: statistical model: If three rams were introduced to a group, and there was no individual hand mating, but group mating, how a specific ram can be included as an effect in the model?
Author’s response#point14: we inserted more information better focusing on mating registration. Rams were provided with marker harnesses so that matings could be recorded every day. Consequently, statistical model remained the same.
Rev1#point15: Line 200: please give in Table 2 the frequency of the different mutation in the 800 samples.
Author’s response#point15: as suggested, we added the frequency of the mutations in all the 800 animals
Rev1#point16: Line 213: Please provide Means and SD for prolificacy of Sarda sheep in this study.
Author’s response#point16: As suggested, we added the information required
Rev1#point17: Tale 4: Please use capital letters for all months and factors
Author’s response#point17: We corrected the table using capital letters
Rev1#point18: Line 224: Rephrase.
Author’s response#point18: we rephrased the sentence hoping is clearer.
Rev1#point19. Line 264: There is no evidence that the two mutations are involved. They are associated….
Author’s response#point19: We change “involved” with “associated”
Rev1#point20: Line 275: The effect on fertility was not significant (Table 3).
Author’s response#point20: we have addressed your concern specifying that the effect on fertility was not significant
Rev1#point21: Line 312: the conclusion is that the polymorphism is associated with variance in fertility. No evidence is for “direct improvement”.
Author’s response#point21: We change the sentence as suggested
Rev1#point22: Line 323: Supporting information is not available on the net.
Author’s response#point22: we are sorry, we forgot to delete this section from the template. Now, thanks to you, we removed it
Reviewer 2 Report
The manuscript “Reproductive resumption in winter and spring related to MTNR1A gene polymorphisms in Sarda sheep” by Mura et al. handles sequencing of exon 2 of the melatonin receptor 1A gene MTNR1A in the Italian sheep breed Sarda. Animals were kept in two commercial farms and grouped and handled in different ways, especially concerning the introduction of a breeding ram. Two polymorphisms were confirmed to have an effect on reproductivity. Especially time between introduction of the ram and lambing was affected by the genotypes/alleles (this is not clear) of both SNP. Interesting correlations were identified, but these are not always presented clearly enough in the current version of the paper. Therefore, I recommend an acceptance of the paper after major revisions.
General concept comments:
First of all be careful to write the abbreviated gene name all the time in italics over the whole manuscript.
The main criticism is that the term “genotype” is, in my opinion, predominantly used incorrectly. The typing called genotypes C/C, C/T and T/T are presumably rather just the individual alleles of the two SNPs? It is not clear whether, for example, the first C refers to the SNP rs430181568 and the second C to the SNP rs407388227. Then it would only be individual alleles of the two SNPs and not genotypes at all, but an allele combination. Or is the C/C the genotype of one or the other SNP? It must always be written which of the two significant SNPs the typing refers to.
Presumably, the designation composite genotype would be more correct and then it would have to be called CC/CC, CC/CT and so on for example, since they are apparently both C/T SNPs. This must be corrected throughout the whole manuscript, otherwise the statements are not comprehensible.
Furthermore, there is a lack of information on allele and genotype (and possibly composite genotype) frequencies over the entire number of animals analysed (n= 800). If it is the case that these are the results of a previous study [26?], this must nevertheless be mentioned here earlier and described in this way.
Specific comments:
Simple summary:
the simple summary contains too much general information and should refer more to the content of the paper.
Line 17: define SNP before using the abbreviation for the first time and write it in capital letters
Line 17: This was already recognised in a previous study and should therefore not be listed here.
Abstract:
Line 23: “..of this genotype…”: which genotype? Here you should give more information: gene name, SNP position, and be carful with the naming “genotype”
Line 27: use “in exon” not “on exon”; MTNR1A in italics!
Line 34: how do you define “fertility”?
Line 36: delete “in position”. The numbers of the SNPs are clear position names.
Keywords: male effect? Why did you choose this key word?
Introduction:
Line 72/73: again here, do not use “genotypes at positions”. You hypothesized that alleles/or genotypes/or composite genotypes? of the SNPs rs430181568 and r2407388227, located IN exon 2 of MTNR1A, are associated with …
Material and Methods
Line 91: delete additional space before “the farms”
Line 97: include “they”. …”where they had…”
Line 99: do not use “subjects”. Use animals or sheep.
Line 101 and later: It is confusing that you already describe the classification of sheep according to their genotypes, although the genotyping is described in the next chapter. Therefore, I would describe the grouping only later. Furthermore, it is not entirely clear why these 240 animals were selected. It was probably because similar groups of animals were formed after the typing. I would describe this as well. Of the 800 animals, 240 were selected on the basis of their alleles/genotypes so that they could be divided into groups of approximately the same age and BCS. If I have understood this correctly?
Table 1: “Group” instead of “groups”; “n =” instead of “n.”, “Genotype or Allele or…” instead of “genotype”
Line 121: who made the scoring? Always the same person?
Line 127: probably 200 µl?
Line 131: as you explained PCR already in line 129 you can use here the abbreviation directly.
Line 135: see comment line 127
Line 138: “dNTP” instead of “dNTPs”
Line 142: why did you choose different annealing temperature for the first 10 and for the last 23 cycles? Please explain as this is not usual.
Line 169: here you write 15th of each month. In line 31 you write 1st of each month. What is correct?
Concerning the rams: each group had how many different rams? If so how could you exclude the effect of the fertility of the ram? Describe rams in more detail (breed? Probably also Sarda?, age, time in breeding production, …) In line 190 you write 12 rams. You should explain this earlier and in more detail.
Results:
Table2: “Identification instead of “identification”, “in the current assemblies…” instead of “on the current assembly…”; “Position in…” instead of “Position on..”; “Alleles?” instead of “Genotypes”
Does SNP rs430181568 also have an effect? Splice sites, transcription factor …? Please make some in silico analyses for this.
Line 202: give a reference to your previous study here, and write that you “confirmed your previous results”.
Table 3: please define the genotypes correctly
Table 4:
“February” instead of “february” or “Februrary”; March and may also capitalised
Discussion:
Line 222: use “exon 2” as you used before.
The discussion is very general. You have to include your results in more detail. The effects of the SNPs on the functionality of the gene and then on fertility need to be described more.
The other 7 SNPs should also be addressed again. Are there possibly also further couplings here? Discuss allele and genotype frequencies.
Why was only exon 2 sequenced and not the whole gene? Missing association as described in Line 222/223 apply for the same animal group? Than this should be explained in the introduction.
More correlations must be made between the literature and your own results of this study. Are there associations between MTNR1A SNPs and fertility in other species for example? It seems that a lot of results are already described in previous papers. The novelty of this research here should be discussed in more detail.
Since the introduction is not particularly long, I would recommend writing some points from the discussion that are described here for the first time in the introduction and instead relating your own results to the literature in the discussion here.
Supplementary material:
As you have no suppl. Material you can delete this point.
Author Contributions:
This should be described in more detail for each author.
Author Response
The manuscript “Reproductive resumption in winter and spring related to MTNR1A gene polymorphisms in Sarda sheep” by Mura et al. handles sequencing of exon 2 of the melatonin receptor 1A gene MTNR1A in the Italian sheep breed Sarda. Animals were kept in two commercial farms and grouped and handled in different ways, especially concerning the introduction of a breeding ram. Two polymorphisms were confirmed to have an effect on reproductivity. Especially time between introduction of the ram and lambing was affected by the genotypes/alleles (this is not clear) of both SNP. Interesting correlations were identified, but these are not always presented clearly enough in the current version of the paper. Therefore, I recommend an acceptance of the paper after major revisions.
Dear reviewer,
Thank you for the revision, your suggestions greatly improved the understanding of our manuscript.
Listed below there are our responses.
General concept comments:
Rev2#point1: First of all be careful to write the abbreviated gene name all the time in italics over the whole manuscript.
Author’s response#point1: Thank you for your suggestion, we changed all the abbreviated gene name in italics
Rev2#point2: The main criticism is that the term “genotype” is, in my opinion, predominantly used incorrectly. The typing called genotypes C/C, C/T and T/T are presumably rather just the individual alleles of the two SNPs? It is not clear whether, for example, the first C refers to the SNP rs430181568 and the second C to the SNP rs407388227. Then it would only be individual alleles of the two SNPs and not genotypes at all, but an allele combination. Or is the C/C the genotype of one or the other SNP? It must always be written which of the two significant SNPs the typing refers to.
Author’s response#point2: we apologize for being unclear. When we used C/C, C/T and T/T we referred to genotypes, not allele combination. The misunderstanding was due to the fact that the SNPs were totally linked (D’=1 and r2=1), therefore, sometime, we have been unclear. Thanks to your note, we revised the manuscript hoping to have modified it addressing your concerns
Rev2#point3: Presumably, the designation composite genotype would be more correct and then it would have to be called CC/CC, CC/CT and so on for example, since they are apparently both C/T SNPs. This must be corrected throughout the whole manuscript, otherwise the statements are not comprehensible.
Author’s response#point3: thank you because this suggestion will improve the comprehension of the manuscript. We changed C/C with CC/CC and other genotypes throughout the whole manuscript as suggested.
Rev2#point4: Furthermore, there is a lack of information on allele and genotype (and possibly composite genotype) frequencies over the entire number of animals analysed (n= 800). If it is the case that these are the results of a previous study [26?], this must nevertheless be mentioned here earlier and described in this way.
Author’s response#point4: In table 2 we added the genotypic and the minor allele frequencies (MAF) considering all the 800 animals genotyped
Specific comments:
Simple summary:
Rev2#point5: the simple summary contains too much general information and should refer more to the content of the paper.
Author’s response#point5:The simple summary has been modified and inserted more content of the study
Rev2#point6: Line 17: define SNP before using the abbreviation for the first time and write it in capital letters
Author’s response#point6: we added "single nucleotide polymorphism" the first time we used the abbreviation
Rev2#point7: Line 17: This was already recognised in a previous study and should therefore not be listed here.
Author’s response#point7: as suggested, we removed the sentence regarding previous studies
Abstract:
Rev2#point8: Line 23: “..of this genotype…”: which genotype? Here you should give more information: gene name, SNP position, and be careful with the naming “genotype”
Author’s response#point8: Thank you for point out the mistake; we replaced “genoype” with “snp” and then, we changed the sentence adding information about snpID and gene.
Rev2#point9: Line 27: use “in exon” not “on exon”; MTNR1A in italics!
Author’s response#point9: we replaced “on” with “in” and, as written in the point 1 of general concept comments, we updated all the abbreviations of genes in italics
Rev2#point10: Line 34: how do you define “fertility”?
Author’s response#point10: We inserted a definition of fertility in order to improve the understanding of the results in the abstract section
Rev2#point11: Line 36: delete “in position”. The numbers of the SNPs are clear position names.
Author’s response#point11: We deleted “in position”
Rev2#point12: Keywords: male effect? Why did you choose this key word?
Author’s response#point12: We think this keyword is important because reproductive activity has been stimulated through this effect. Therefore, in our opinion, future researchers will be able to use these findings to support their work if they conduct studies in which the male impact is present.
Rev2#point13: Line 72/73: again here, do not use “genotypes at positions”. You hypothesized that alleles/or genotypes/or composite genotypes? of the SNPs rs430181568 and r2407388227, located IN exon 2 of MTNR1A, are associated with …
Author’s response#point13: Sorry for the repetitive mistake, we updated the sentence following your suggestions
Material and Methods
Rev2#point14: Line 91: delete additional space before “the farms”
Author’s response#point14: We delete the space
Rev2#point15: Line 97: include “they”. …”where they had…”
Author’s response#point15: we added “they”
Rev2#point16: Line 99: do not use “subjects”. Use animals or sheep.
Author’s response#point16: We replaced “subjects” with “sheep”
Rev2#point17: Line 101 and later: It is confusing that you already describe the classification of sheep according to their genotypes, although the genotyping is described in the next chapter. Therefore, I would describe the grouping only later. Furthermore, it is not entirely clear why these 240 animals were selected. It was probably because similar groups of animals were formed after the typing. I would describe this as well. Of the 800 animals, 240 were selected on the basis of their alleles/genotypes so that they could be divided into groups of approximately the same age and BCS. If I have understood this correctly?
Author’s response#point17: you understood correctly, and we added your suggested sentence. Thanks to your correction the understanding of the manuscript is improved
Rev2#point18: Table 1: “Group” instead of “groups”; “n =” instead of “n.”, “Genotype or Allele or…” instead of “genotype”
Author’s response#point18: as suggested by you, we replaced throughout the manuscript C/C, C/T and T/T with CC/CC, CT/CT and TT/TT. Furthermore, we made the correction required
Rev2#point19. Line 121: who made the scoring? Always the same person?
Author’s response#point19: yes, the same person, so we added this information
Rev2#point20: Line 127: probably 200 µl?
Author’s response#point20: yes, we added “µ”
Rev2#point21: Line 131: as you explained PCR already in line 129 you can use here the abbreviation directly.
Author’s response#point21: we deleted the explanation
Rev2#point22: Line 135: see comment line 127
Author’s response#point22: yes, we added “µ”
Rev2#point23: Line 138: “dNTP” instead of “dNTPs”
Author’s response#point23: we deleted the “s”
Rev2#point24: Line 142: why did you choose different annealing temperature for the first 10 and for the last 23 cycles? Please explain as this is not usual.
Author’s response#point24: thank you so much for point out the mistake. Some PCR conditions reported in the text were wrong. We replaced with the correct version.
Rev2#point25: Line 169: here you write 15th of each month. In line 31 you write 1st of each month. What is correct?
Author’s response#point25: thank you for pointing out the mistake; the correct date was the 15th, consequently, we corrected all the dates.
Rev2#point26: Concerning the rams: each group had how many different rams? If so how could you exclude the effect of the fertility of the ram? Describe rams in more detail (breed? Probably also Sarda?, age, time in breeding production, …) In line 190 you write 12 rams. You should explain this earlier and in more detail.
Author’s response#point26: we added information about rams “The rams (Sarda breed) were four years old, and they have already been produced offspring in the previous mating seasons. Furthermore, a veterinarian performed the genitals examination and did not find any abnormalities of the reproductive system.”
Results:
Rev2#point27. Table2: “Identification instead of “identification”, “in the current assemblies…” instead of “on the current assembly…”; “Position in…” instead of “Position on..”; “Alleles?” instead of “Genotypes”
Does SNP rs430181568 also have an effect? Splice sites, transcription factor …? Please make some in silico analyses for this.
Author’s response#point27: we updated the table as suggested. All the mistakes have been corrected. Surely, this method of analysis could clarify which of the SNPs has the greatest influence in regulating reproductive activity. The idea is very good, and thank you for that, but we are unable to perform this analysis at the moment. In future studies we will try to improve our knowledge and we will certainly include this analysis as well.
Rev2#point28: Line 202: give a reference to your previous study here, and write that you “confirmed your previous results”.
Author’s response#point28: we added here the reference
Rev2#point29: Table 3: please define the genotypes correctly
Author’s response#point29: as suggested, we replaced throughout the manuscript C/C, C/T and T/T with CC/CC, CT/CT and TT/TT. We made the same in this table
Rev2#point30: Table 4: “February” instead of “february” or “Februrary”; March and may also capitalised
Author’s response#point30: all the moths have been capitalized
Discussion:
Rev2#point31: Line 222: use “exon 2” as you used before.
Author’s response#point31: We replace “II exon” with ”exon 2”
Rev2#point32: The discussion is very general. You have to include your results in more detail. The effects of the SNPs on the functionality of the gene and then on fertility need to be described more.
Author’s response#point32: Several sentences have been added to the discussion to better clarify the effect of different genotypes on reproductive activity.
Rev2#point33: The other 7 SNPs should also be addressed again. Are there possibly also further couplings here? Discuss allele and genotype frequencies.
Author’s response#point33: In our paper on theriogenology (158, 438-444, 2020) we have extensively discussed the other SNPs and it seems redundant to discuss them again. However, following your suggestion we have inserted a sentence where we refer to that publication also mentioned in the introduction
Rev2#point34: Why was only exon 2 sequenced and not the whole gene? Missing association as described in Line 222/223 apply for the same animal group? Than this should be explained in the introduction.
Author’s response#point34: In the introduction section, the following sentence was added to explain why we only studied exon 2 “In our previous studies on the Sarda breed different SNPs were found in exon 1 and 2 but only those mentioned above in exon 2 showed association with reproductive activity”.
Rev2#point35: More correlations must be made between the literature and your own results of this study. Are there associations between MTNR1A SNPs and fertility in other species for example? It seems that a lot of results are already described in previous papers. The novelty of this research here should be discussed in more detail.
Author’s response#point35: As for the effect of the polymorphisms of the MTNR1A gene on reproductive activity, certainly it is a very important topic but it seems very dispersive in our opinion as the polymorphisms are very different from those of the sheep. We believe that to avoid confusion and to simplify the understanding of the manuscript, it is better not to include this topic. In European goat, for example, SNPs have less impact on reproductive seasonality than in Chinese goats, as we have also indicated (Carcangiu et al., 2009). Another specie that presented associations between this gene and reproductive activity is the buffalo and we were the first to report this feature (Carcangiu et al., Theriogenology2011, 76,419-426, Luridiana et al., Reproduction Fertility and Development, 24,983-987 ).
Rev2#point36: Since the introduction is not particularly long, I would recommend writing some points from the discussion that are described here for the first time in the introduction and instead relating your own results to the literature in the discussion here.
Author’s response#point36: Following your suggestion, some sentences were cut from the introduction and carried over to the discussion to better explain the importance of the results
Rev2#point37: Supplementary material: As you have no suppl. Material you can delete this point.
Author’s response#point37: we are sorry, we forgot to delete this section from the template. Thank you, we removed it
Rev2#point38: Author Contributions: This should be described in more detail for each author.
Author’s response#point38: we described with more details author contributions
Reviewer 3 Report
This manuscript reports an association study between two single nucleotide polymorphisms, rs430181568 and rs407388227, located on the exon 2 of the MTNR1A and the reproductive recovery of Sarda sheep in different months of ram introduction in the flock. The paper is well written. Results contribute to the literature on reproductive function and show an interest in breeding.
Main concerns:
Although authors state that ewes are from two sheep farms, what are the genetic relationship or origins of the ewes? From different families? How were the 240 ewes chosen? At random, as unrelated as possible, based on pedigree records, others? Please elaborate more!. This is an import point because the ewes could be related and some structure of the population could exist, and then the association analyses could be influenced. The pedigree should be included in the analyses.
In Table 1 authors should include the standard deviation of average age and BCS to show that the selected ewes were homogeneous based on these traits. In the association studies, age and BCS should be included.
Authors sequence the whole exon 2. Why the authors have not included the results of the association studies for the other SNPs showed in Table 2? MAF should be included. Furthermore, an haplotype association study using all exon 2 SNPs could be very informative, and the authors could confirm their results and find other SNPs influencing the studied trait.
Author Response
This manuscript reports an association study between two single nucleotide polymorphisms, rs430181568 and rs407388227, located on the exon 2 of the MTNR1A and the reproductive recovery of Sarda sheep in different months of ram introduction in the flock. The paper is well written. Results contribute to the literature on reproductive function and show an interest in breeding.
Dear reviewer,
Thank you for the revision, your suggestions greatly improved the understanding of our manuscript.
Listed below there are our responses.
Main concerns:
Rev3#point1: Although authors state that ewes are from two sheep farms, what are the genetic relationship or origins of the ewes? From different families? How were the 240 ewes chosen? At random, as unrelated as possible, based on pedigree records, others? Please elaborate more! This is an import point because the ewes could be related and some structure of the population could exist, and then the association analyses could be influenced. The pedigree should be included in the analyses.
Author’s response#point1: thank you for your suggestions because you give us the opportunity to improve the understanding of the manuscript; we genotyped 800 animals (400 from each farm) among them, 240 (120 from each farm) were selected on the basis of their genotypes for the SNPs rs430181568 and rs407388227, so that they could be divided into groups of approximately the same age and BCS. Our idea was to make homogeneous groups with little differences among the animals selected. Pedigree in this study has not been included as in the common Sarda sheep farming system once males are introduced with ewes, mating happens randomly. Thus, the origin of the offspring, is usually unknown. Thank you for your suggestion that it could be an idea for a further study.
Rev3#point2: In Table 1 authors should include the standard deviation of average age and BCS to show that the selected ewes were homogeneous based on these traits. In the association studies, age and BCS should be included.
Author’s response#point2: Yes, standard deviation for age and BCS has been added in Table 1.
Rev3#point3: Authors sequence the whole exon 2. Why the authors have not included the results of the association studies for the other SNPs showed in Table 2? MAF should be included. Furthermore, an haplotype association study using all exon 2 SNPs could be very informative, and the authors could confirm their results and find other SNPs influencing the studied trait.
Author’s response#point3: As suggested, we added MAF in a column of the table. Instead, the haplotype association study using all exon 2 SNPs has already been published by our research group (cited with number 24 in the manuscript) so, in this study, we focused our attention on those SNPs associated with reproductive activity (New polymorphisms at MTNR1A gene and their association with reproductive resumption in Sarda breed sheep. Luridiana, S., Cosso, G., Pulinas, L., di Stefano, M. V., Curone, G., Carcangiu, V., Mura, M. C. (2020). Theriogenology, 158, 438–444. https://doi.org/10.1016/j.theriogenology.2020.10.006). In the same paper we used Haplostats of R Statistical software to analyse the haplotypes discovered.
Reviewer 4 Report
This is an interesting manuscript aimed to study the effects of two polymorphisms from the MTNR1A gene on reproductive traits in Sarda sheep. However, there are some major concerns about the methodology, results and conclusions that must be taken in account.
General comments:
- Marker SNPs have to meet at least these 2 quality requirements in order to be considered for an associative study: Hardy-Weinberg Equilibrium test (HWE, X2 > 0.05) and minor allele frequency higher than 5 or 10% (MAF > 0.05 or 0.10). None of these 2 tests were included in the methodology.
- Table 2 and Table 3 must be separated according to each SNP, because only in this way is possible to analyze results for concluding that the polymorphisms rs430181568 and rs407388227 of the MTNR1A gene were able to improve both fertility rate and DRIL. I suggest to run the statistical model by separate to analyze the effect of each individual SNP.
- Authors assumed that the 2 mutations in position rs430181568 and rs407388227 were totally linked, but no test was included in the study to prove this. A linkage disequilibrium (LD) test is needed to confirm the 2 mutations are linked, in order to be considered as as a single marker piece (i.e., haplotype block).
- I suggest to improve the discussion by including some references to explain the effect of the 2 SNPs on litter size (no significant) and DRIL (significant).
- In References section, all references should be corrected following the guidelines described in the “Instructions for Authors”.
Author Response
This is an interesting manuscript aimed to study the effects of two polymorphisms from the MTNR1A gene on reproductive traits in Sarda sheep. However, there are some major concerns about the methodology, results and conclusions that must be taken in account.
Dear reviewer,
Thank you for the revision, your suggestions greatly improved the understanding of our manuscript.
Listed below there are our responses.
General comments:
Rev4#point1: Marker SNPs have to meet at least these 2 quality requirements in order to be considered for an associative study: Hardy-Weinberg Equilibrium test (HWE, X2 > 0.05) and minor allele frequency higher than 5 or 10% (MAF > 0.05 or 0.10). None of these 2 tests were included in the methodology.
Author’s response#point1: thank you for pointing out this lack. We agreed with you and we added Hardy Weinberg values and MAF in Table 2
Rev4#point Table 2 and Table 3 must be separated according to each SNP, because only in this way is possible to analyze results for concluding that the polymorphisms rs430181568 and rs407388227 of the MTNR1A gene were able to improve both fertility rate and DRIL. I suggest to run the statistical model by separate to analyze the effect of each individual SNP.
Author’s response#point2: Thank you for your suggestions, we modified the text after to have performed a linkage disequilibrium test (as suggested by you in the next point). After this confirmation, we added the values of D’ and r2 (D’=1 and r2=1), so the results of the statistical model are the same.
Rev4#point Authors assumed that the 2 mutations in position rs430181568 and rs407388227 were totally linked, but no test was included in the study to prove this. A linkage disequilibrium (LD) test is needed to confirm the 2 mutations are linked, in order to be considered as a single marker piece (i.e., haplotype block).
(D’=1 andr2=1)
Author’s response#point3: as specified in the previous point, we updated the manuscript as suggested. Thank you because this suggestion made the manuscript clearer.
I suggest to improve the discussion by including some references to explain the effect of the 2 SNPs on litter size (no significant) and DRIL (significant).
Author’s response#point4: thank you for the suggestion, in discussion we added more information about litter size and DRIL
Rev4#point In References section, all references should be corrected following the guidelines described in the “Instructions for Authors”.
Author’s response#point5: We updated the reference section following the “Instructions for Authors”.
Round 2
Reviewer 1 Report
Review 1st revision: Reproductive resumption in winter and spring related to 2 MTNR1A gene polymorphisms in Sarda sheep. Animals 2022.
Reading the revised manuscript brought serious reservation regarding the experimental design, the statistical analysis and the interpretation of the results.
Experimental design:
1. In the introduction (lines 78-80) the authors state that “…the aim of the present study is to verify if the animals carrying the different genotypes of these two polymorphisms show a different reproductive recovery after the photorefractoriness period (February and May)”. However, what was studied was response to a “ram effect” on resumption of reproductive activity in “out of season” period. Those are two, different traits. To study natural reproductive activity one needs to monitor hormonal ovarian activity in absence of rams.
2. To set up the experiment, 120 ewes from one farm were transferred to a second one. It is not stated when the transfer and forming the experimental groups took place prior February, when the rams were introduced for the first time. Moving animals to a new location may affect their reproductive behavior.
3. I was stated in the authors’ reply: Each group was housed in separate paddocks distant approximately 4Km from each other, in order to avoid visual, olfactory, and sound signals. We added this additional information in the text. This should be declared in the text.
4. It is not clear how the rams were distributed between groups. If there were 6 rams from each farm, in a group (3 rams) 2 where from one farm and one from the other. How this difference is treated?
5. It is not clear how the sire of the lambs were identified and “a ram effect was included in the statistical analysis. Actually, this was a “Group mating”. Did each ram carry crayon with a different color? How cases of double-marking where addressed. How cases of no marking where addressed?
6. Having results for all ewes on a list of SNPs at the MTNR1A gene (Table 2), the experimental design and the statistical analysis had to be plan from the beginning to test the association between all those SNPs and the traits in study. Previous information on association of other traits with SNPs is not relevaqnt.
Statistical model:
1. The “farm” effect should be included both for ewes and rams effects. It is stated that (line 201): The fixed effect of the farm was tested but once it resulted not statistically significant we removed from the model. This is a mistake. The Analysis has to include the farm effect and the results of this analysis have to be presented. One may remove the non-significant effects only for calculating LSM.
2. The ram effect has to be nested within group. It is not a random effect rams allocated to one group could not serve in principle in other groups.
Results and interpretation of the results:
1. If there is an effect, it is not clear whether it is genotype differences in response to a ram effect or different photoperiod activity.
2. No information is given on % of abortions and returning on heat in the experimental population.
3. In many places in the manuscript (see for example line 26: The aim of the present research was to evaluate the effect ...). This is a mistake: The study looks for association and not for an effect.
4. The authors should address the possibility of selecting for the “positive” alleles, according to their findings, as a mean to improve Sarda reproductive performance.
Author Response
Dear reviewer,
Thank you for the revision, your suggestions greatly improved the understanding of our manuscript.
Listed below there are our responses.
Rev1#point1: In the introduction (lines 78-80) the authors state that “…the aim of the present study is to verify if the animals carrying the different genotypes of these two polymorphisms show a different reproductive recovery after the photorefractoriness period (February and May)”. However, what was studied was response to a “ram effect” on resumption of reproductive activity in “out of season” period. Those are two, different traits. To study natural reproductive activity one needs to monitor hormonal ovarian activity in absence of rams.
Author’s response#point1: Thank you for the suggestion, certainly, as you said, they are two different traits: the male effect stimulates reproductive activity in anestrus animals whilst induces the mating in those in estrus. Therefore, in order to improve the accuracy of the experiment it would have been necessary to make the dosage of several hormones to monitor the resumption of reproductive activity. However, the monitoring the reproductive recovery indirectly, through the male effect, can give a great indication of the reproductive status of the animals. Furthermore, performing hormone analysis in a wide number of animals is expensive and we didn’t have the financial resources to make them. Surely, in subsequent research we will consider your suggestion by inserting hormonal control, perhaps decreasing the number of animals.
Rev1#point2: To set up the experiment, 120 ewes from one farm were transferred to a second one. It is not stated when the transfer and forming the experimental groups took place prior February, when the rams were introduced for the first time. Moving animals to a new location may affect their reproductive behavior.
Author’s response#point2: thanks, because you gave the chance to improve the understanding of the manuscript. In order to better explain the experimental design, we have included some other information about the two farms involved in the study. As mentioned in the manuscript, the two farms are located in nearby area and have adjacent pastures, so they have the same climatic situation and the farm management is similar (the same nutritionist and veterinarian follow the two farms). 30 days before the start of the experiment (January 15th), the 4 groups were formed and kept in one farm, so that the animals could adapt to the new life situation. The date of group formation was better specified in the text.
Rev1#point3: I was stated in the authors’ reply: Each group was housed in separate paddocks distant approximately 4Km from each other, in order to avoid visual, olfactory, and sound signals. We added this additional information in the text. This should be declared in the text.
Author’s response#point3: Thank you for the suggestion, we added this information in the text
Rev1#point4: It is not clear how the rams were distributed between groups. If there were 6 rams from each farm, in a group (3 rams) 2 where from one farm and one from the other. How this difference is treated?
Author’s response#point4: The males of the two farms were kept in one group in order to uniform the management of the males. In the previous points, we said that 4 groups were formed and, therefore, 3 males were introduced into each group of 60 animals. Therefore, the rams received the same management. This information has been inserted into the text.
Rev1#point5: It is not clear how the sire of the lambs were identified and “a ram effect was included in the statistical analysis. Actually, this was a “Group mating”. Did each ram carry crayon with a different color? How cases of double-marking where addressed. How cases of no marking where addressed?
Author’s response#point5: we agree with you and we updated the model following your suggestions. The males introduced into the group had the same colour marker and did not give the possibility of identifying which ram a specific ewe mated with. Therefore, no paternity has been established. Therefore, as suggested by you, we removed the ram effect from the statistical model. Each group mating corresponded to each experimental group.
Rev1#point6: Having results for all ewes on a list of SNPs at the MTNR1A gene (Table 2), the experimental design and the statistical analysis had to be plan from the beginning to test the association between all those SNPs and the traits in study. Previous information on association of other traits with SNPs is not relevant.
Author’s response#point6: at the beginning of the study, we planned to analyse all the SNPs in order to evaluate the possible associations with reproductive traits, as indeed you suggested. However, in other study we have extensively analysed the association among SNPs in MTNR1A gene and the same reproductive traits, so it seems redundant to report them again.
Rev1#point7: The “farm” effect should be included both for ewes and rams effects. It is stated that (line 201): The fixed effect of the farm was tested but once it resulted not statistically significant we removed from the model. This is a mistake. The Analysis has to include the farm effect and the results of this analysis have to be presented. One may remove the non-significant effects only for calculating LSM.
Author’s response#point7: in the text we made a mistake by inserting the effect farm because the animals came from two farms. In fact, it was superfluous to insert this variable as the animals were grouped into a single group. The groups were formed a month earlier and therefore the farm effect should not be taken into consideration. Therefore, we have eliminated the effect from the model.
Rev1#point8: The ram effect has to be nested within group. It is not a random effect rams allocated to one group could not serve in principle in other groups.
Author’s response#point8: following your previous suggestions we updated the model and we removed the ram effect.
Results and interpretation of the results:
Rev1#point9: If there is an effect, it is not clear whether it is genotype differences in response to a ram effect or different photoperiod activity.
Author’s response#point9: The association of genotypes and reproductive recovery is evident as reported in the tables and described in the manuscript. Animals with the CC/CC and CT/CT genotypes show a better reproductive recovery after the male effect than TT/TT animals. This fact shows that the two genotypes are associated with reproductive recovery.
Rev1#point10: No information is given on % of abortions and returning on heat in the experimental population.
Author’s response#point10: we agree and have added these data. There were no abortions detected amongst any of the ewes. In groups A and B, no return to estrus cycle was recorded, while in groups C and D, 2 and 3 ewes. As required we added this information in the text
.
Rev1#point11: In many places in the manuscript (see for example line 26: The aim of the present research was to evaluate the effect ...). This is a mistake: The study looks for association and not for an effect.
Author’s response#point11: this observation is correct, we made the changes throughout the manuscript
Rev1#point12: The authors should address the possibility of selecting for the “positive” alleles, according to their findings, as a mean to improve Sarda reproductive performance.
Author’s response#point12: in the conclusion section, we added this information
Reviewer 2 Report
In the revised form of the manuscript “Reproductive resumption in winter and spring related to MTNR1A gene polymorphisms in Sarda sheep” by Mura et al. most of my comments were satisfactorily edited. The corrected manuscript is significantly more comprehensible and much improved. I only noticed a few minor points, so I recommend acceptance after minor revision.
Abstract:
Line 25: probably you mean “20 ewes carrying CC/CC, 20 CT/CT and 20 TT/TT genotype”, as also described later in this way (line 108/109 for example)?
Material and Methods
Line 107/108: for me these details of the allele frequencies are results and should be mentioned in the results chapter
Line 110: I do not understand this sentence. 240 animals grouped in one farm but 120 animals from each farm?? Are 120 animals from each farm selected and afterwards kept in one farm? This is not really understandable in this way.
Line 139: additional space after “µl”. Please delete.
Line 170: two points “..” before “Before…”.
Line 176: What is the meaning of MMC1?
Lime 182: What is the meaning of MMC2? And insert a “.” after MMC2.
Results:
Line 210: “genotype frequencies” instead of “genotypes frequency”.
Line 212: What is the meaning of SIFT?
Table 4: delete “r” in “Februrary”
Discussion:
Line 274 and 284: insert a “,” in the cited literature bracket.

Author Response
Dear reviewer,
Thank you for the revision, your suggestions greatly improved the understanding of our manuscript.
Listed below there are our responses.
In the revised form of the manuscript “Reproductive resumption in winter and spring related to MTNR1A gene polymorphisms in Sarda sheep” by Mura et al. most of my comments were satisfactorily edited. The corrected manuscript is significantly more comprehensible and much improved. I only noticed a few minor points, so I recommend acceptance after minor revision.
Abstract:
Rev2#point1: Line 25: probably you mean “20 ewes carrying CC/CC, 20 CT/CT and 20 TT/TT genotype”, as also described later in this way (line 108/109 for example)?
Author’s response#point1: Thank you for pointing our mistake out, we changed CC/TT with CT/CT
Material and Methods
Rev2#point2: Line 107/108: for me these details of the allele frequencies are results and should be mentioned in the results chapter
Author’s response#point2: we also wanted to include these two data in the results, as you suggested, but it seemed incorrect. In fact, these two values were calculated to form the experimental groups and therefore it seemed logical to include them in the materials and methods and not repeat them in the results.
Rev2#point3: Line 110: I do not understand this sentence. 240 animals grouped in one farm but 120 animals from each farm?? Are 120 animals from each farm selected and afterwards kept in one farm? This is not really understandable in this way.
Author’s response#point3: we changed the sentence in order to improve the understanding
Rev2#point4: Line 139: additional space after “µl”. Please delete.
Author’s response#point4: we removed the additional space
Rev2#point5: Line 170: two points “..” before “Before…”.
Author’s response#point5: we removed the additional “.”
Rev2#point6: Line 176: What is the meaning of MMC1?
Author’s response#point6: Sorry, we had forgotten to remove one comment, in the current version we deleted it
Rev2#point7: Lime 182: What is the meaning of MMC2? And insert a “.” after MMC2.
Author’s response#point7: Thanks again, as said in the previous point it was a comment, now we removed “MMC2” and inserted the “.”
Results:
Rev2#point8: Line 210: “genotype frequencies” instead of “genotypes frequency”.
Author’s response#point8: we changed as suggested
Rev2#point9: Line 212: What is the meaning of SIFT?
Author’s response#point9: we added the meaning of “SIFT”
Rev2#point10: Table 4: delete “r” in “Februrary”
Author’s response#point10: thanks, we corrected “Februrary” with “February”
Discussion:
Rev2#point11: Line 274 and 284: insert a “,” in the cited literature bracket.
Author’s response#point11: We added the commas
Reviewer 3 Report
The authors have satisfactorily addressed most of my concerns. However, the structure of the population is an important concern. I understand the farming system, and that there are not records about the putative fathers because of the reproductive management. Furthermore, the analysed SNPs are not in HW equilibrium. Association studies using SNPs that are not in HW equilibrium should be taken with caution. The HW disequilibrium could be because of the structure of the population, that have not been taken into account in this study. The authors may include in the model some of the pedigree information known to try to solve this concern, for example mothers.
Finally, authors have included standard deviation for age and BCS. BCS should be included in the model. I see that the BCS are similar with low sd, but this is an important factor that could affect reproductive resumption trait.
Author Response
Dear reviewer,
Thank you for the revision, your suggestions greatly improved the understanding of our manuscript.
Listed below there are our responses.
Rev3#point1: The authors have satisfactorily addressed most of my concerns. However, the structure of the population is an important concern. I understand the farming system, and that there are not records about the putative fathers because of the reproductive management. Furthermore, the analysed SNPs are not in HW equilibrium. Association studies using SNPs that are not in HW equilibrium should be taken with caution. The HW disequilibrium could be because of the structure of the population, that have not been taken into account in this study. The authors may include in the model some of the pedigree information known to try to solve this concern, for example mothers.
Author’s response#point1:
We agree with you, but these data are not in our possession. In Sardinia 3,200,000 sheep are raised and about 300,000 sheep are registered in the herd book. Therefore, farms that are not registered in the herd book do not register the paternity and maternity of the animals. We conducted the experimentation in two non-registered farms and therefore did not have the animal genealogy records. Regarding allelic and genotypic frequencies, in this study we found the same frequencies our previous published research. On the other hand, we have genotyped several thousand sheep, from different areas of Sardinia, and we find approximately the same frequencies.
Rev3#point2: Finally, authors have included standard deviation for age and BCS. BCS should be included in the model. I see that the BCS are similar with low sd, but this is an important factor that could affect reproductive resumption trait.
Author’s response#point2: thank you for giving the chance to improve the manuscript. The suggestion has been added in the statistical model
Reviewer 4 Report
The manuscript has improved significantly; however, I suggest to consider next two minor comments before proceed for publication:
- Please briefly describe in “Materials and Methods” section the procedures used to calculate minor allele frequency (MAF) and to test for Hardy-Weinberg Equilibrium (HWE).
- In “References” section the name of the authors should follow the guidelines described in the “Instructions for Authors”. Please correct all references.
Author Response
Dear reviewer,
Thank you for the revision, your suggestions greatly improved the understanding of our manuscript.
Listed below there are our responses.
The manuscript has improved significantly; however, I suggest to consider next two minor comments before proceed for publication:
Rev4#point1: Please briefly describe in “Materials and Methods” section the procedures used to calculate minor allele frequency (MAF) and to test for Hardy-Weinberg Equilibrium (HWE).
Author’s response#point1: Thanks, as suggested we added the procedures used
Rev4#point2: In “References” section the name of the authors should follow the guidelines described in the “Instructions for Authors”. Please correct all references.
Author’s response#point2: Thank you for pointing the mistake out, we changed all the references following “Instructions for Authors”